# Imputing pre-diagnosis health behaviour in cancer registry data and investigating its relationship with oesophageal cancer survival time

**Paul P. Fahey** [1]*, **Andrew Page**[1], **Thomas Astell-Burt**[2], **Glenn Stone**[3]

1 Translational Health Research Institute, Western Sydney University, Campbelltown, New South Wales, Australia, 2 Population Wellbeing and Environment Research Lab (PowerLab), School of Health and Society, Faculty of Social Sciences, University of Wollongong, Wollongong, New South Wales, Australia, 3 Glenn Stone, School of Computer, Data and Mathematical Sciences, Western Sydney University, Parramatta, New South Wales, Australia

* p.fahey@westernsydney.edu.au

**Data Availability Statement:** The data underlying the results presented in the study are available from the US National Cancer Institute (https://seer.cancer.gov/data/) and the US Centers for Disease

## Abstract

### Background

As oesophageal cancer has short survival, it is likely pre-diagnosis health behaviours will have carry-over effects on post-diagnosis survival times. Cancer registry data sets do not usually contain pre-diagnosis health behaviours and so need to be augmented with data from external health surveys. A new algorithm is introduced and tested to augment cancer registries with external data when one-to-one data linkage is not available.

### Methods

The algorithm is to use external health survey data to impute pre-diagnosis health behaviour for cancer patients, estimate misclassification errors in these imputed values and then fit misclassification corrected Cox regression to quantify the association between pre-diagnosis health behaviour and post-diagnosis survival. Data from US cancer registries and a US national health survey are used in testing the algorithm.

### Results

It is demonstrated that the algorithm works effectively on simulated smoking data when there is no age confounding. But age confounding does exist (risk of death increases with age and most health behaviours change with age) and interferes with the performance of the algorithm. The estimate of the hazard ratio (HR) of pre-diagnosis smoking was HR = 1.32 (95% CI 0.82,2.68) with HR = 1.93 (95% CI 1.08,7.07) in the squamous cell sub-group and pre-diagnosis physical activity was protective of survival with HR = 0.25 (95% CI 0.03, 0.81). But the method failed for less common behaviours (such as heavy drinking).

Control and Prevention (https://www.cdc.gov/brfss/data_documentation/index.htm).

**Funding:** The author(s) received no specific funding for this work.

**Competing interests:** The authors have declared that no competing interests exist.

## Conclusions

Further improvements in the I2C2 algorithm will permit enrichment of cancer registry data through imputation of new variables with negligible risk to patient confidentiality, opening new research opportunities in cancer epidemiology.

## Introduction

Oesophageal cancer is an important cancer. Worldwide it is estimated that it accounts for 3.1% of all cancers and 5.5% of all cancer deaths [1]. But it is still a relatively rare disease. In the US for example, the estimated risk of being diagnosed with oesophageal cancer before age 75 is just 1.15% in males and 0.44% in females [1]. There were an estimated 184,400 new cases (4.3 per 100,000 population) in the US in 2020 with an estimated 5-year relative survival of just under 20% [2].

Given the relatively short survival time, it is likely that pre-diagnosis health behaviour is an important contributor to post-diagnosis survival time. It is important to document this relationship to help understand how current changes in health behaviour (such as decreasing smoking rates [3], changes in alcohol consumption patterns [4, 5], increasing leisure time physical activity [6] and increasing obesity rate [7]) will impact on the future number of oesophageal cancer survivors and their associated health service needs. Also, at the patient level, clearer understanding of the effect of pre-diagnosis health behaviour may assist in addressing the current weaknesses in prognostic indexes for oesophageal cancer [8].

Cancer registries provide high quality, census data for cancers and often record patient outcomes such as survival. Unfortunately, cancer registries rarely contain pre-diagnosis behavioural risk factors. Retrospective data collection (contacting cases from the cancer registry and interviewing them about their pre-diagnosis behaviours) are subject to survival and recall biases and data collection costs. An alternative approach is to augment the cancer registry data through record linkage with an external data source. But locating routinely collected pre-diagnosis records of oesophageal cancer patients' health behaviour is challenging. While regular population-based surveys of health behaviour are conducted across a range of settings, the rarity of oesophageal cancer (4.3 people per 100,000 per year in the US [2]), means that very few survey respondents would have gone on to experience oesophageal cancer. Individually linked data sets will be small and, with a low proportion of correct links, and linkage errors may dominate.

We have recently described an algorithm [9], which we call the I2C2 (Impute, Impute, Calibrate, Correct) approach, to investigate the relationship between health behaviour and relative risk of surviving 12 months after diagnosis. This does not require cancer registry cases to be present in the health survey data set. It only requires survey participants to be demographically similar to those with data in the cancer registry. This not only avoids the small sample sizes for true matches, but also avoids issues of confidentiality and costs associated with data linkage.

The results of the previous application of this approach [9] displayed sufficient face-validity to extend the algorithm to the estimation of hazard ratios through Cox regression. Accordingly, the aim of this paper is to describe the relationship between pre-diagnosis health behaviour and post-diagnosis cancer survival by augmenting a cancer registry data set with information from an external, no-cases-in-common, health behaviour data set.

## Methods

The project was approved by the Western Sydney University Human Research Ethics Committee (H12305).

### Data sources

Cancer registry data were obtained from the US Surveillance, Epidemiology, and End Results Program (SEER) cancer registries data base (https://seer.cancer.gov/data/). The SEER data base is compiled annually and collates data from US cancer registries with a combined coverage of approximately 28% of the US population [10]. Data from all 39,233 cases of primary malignant oesophageal cancer for the 10 most recent years (2006–2015 at time of data access) were extracted. The 123 cancer cases aged less than 35 years were excluded as atypical.

Reference data on health behaviours, used to help impute health behaviour for the SEER cancer cases, were obtained from the Behavioural Risk Factor Surveillance System (BRFSS) (https://www.cdc.gov/brfss/data_documentation/index.htm). This is an annual telephone survey conducted in each US State and Territory, compiling behavioural data for more than 400,000 adult residents per year [11]. To meet the criteria of 'pre-diagnosis' BRFSS data records from 5-years prior to cancer diagnosis were used. That is, the 3,469,905 BRFSS health survey data records from 2001 to 2010 were included. But the 2,515,009 data records from residents of US States outside the SEER population catchments were excluded, leaving 954,896 records. (With an estimated incidence of 4.3 per 100,000 [2] just 41 of these 954,896 health survey respondents could be expected to be diagnoses with oesophageal cancer in any given year.)

### Variables

The outcome variable was post-diagnosis all cause survival time measured in months. Individuals who survived beyond 2015 and those who were lost to follow-up had their censored survival time recorded at their last known survival date. The maximum possible follow-up time was 119 months. SEER cancer registry records with missing survival time (n = 512) were excluded from the analysis.

Self-reported health behaviour variables were selected from the BRFSS health survey, and included:

- Current tobacco smoking (yes or no), defined as daily or less than daily smoking;

- Alcohol consumption–possible binge drinking (yes or no), defined as ≥5 standard drinks for males or ≥4 standard drinks for females on at least one occasion in the month prior to survey;

- Alcohol consumption–possible heavy drinking (yes or no), defined as >2 standard drinks per day for men and >1 standard drink per day for women in the month prior to survey;

- Physical activity (yes or no), defined as any physical activity or exercise in the past 30 days other than for regular job;

- Obese (yes/no), defined as body mass index ≥ 30 kg/m$^2$; and

- Current tobacco smoking with regular alcohol (yes or no), defined as current tobacco smoking with ≥1 standard drink of alcohol per day on average in the previous month.

The SEER cancer registry variables used to inform the imputation process were:

- Age category at diagnosis (5-year groups from 35-39y to 75-79y then >80y);

- Gender (male; female);

- Marital status (married, including common law; single or never married; widowed; divorced);

- Race (white; black; Asian or Pacific Islander; American Indian or Alaska Native);

- State of residence (Alaska; California; Connecticut; Georgia; Hawaii; Iowa; Kentucky; Louisiana; Michigan; New Jersey; New Mexico; Utah; Washington);

- Year of diagnosis (2006 to 2015).

The same variables were extracted from the BRFSS health survey data sets from 2001 to 2010. The BRFSS health survey records were from 5 years earlier than the SEER cancer cases so as to correspond to 'pre-diagnosis' behaviour. SEER cancer registry data records (n = 2,344, 6.0%) and BRFSS health survey data records (n = 18,770, 2.0%) with missing data on any of these variables were excluded from further analyses.

Sub-group analyses were conducted on adenocarcinoma and squamous cell carcinoma as different health behaviours may have different impacts on the two types of oesophageal cancer [12].

## The I2C2 algorithm

Conceptually, first visualise the pre-diagnosis health behaviours as variables in the SEER cancer registry with 100% missing data. Perhaps these missing data could be addressed by imputation. As health behaviour is missing for all SEER cancer registry records, any such imputation requires external data on health behaviour: in this case the BRFSS health survey data set.

In previous publications we have described two different approaches to imputing SEER cancer cases' behaviour from BRFSS health survey data via common demographic variables. One approach was to develop a logistic regression model predicting behaviour from demographic variables in the BRFSS health survey data and then apply that model to each SEER cancer case to estimate their probability of having the behaviour [13]. The second approach was to stratify both data sets into age by sex by race by marital status by State of residence by year subgroups, and then within each strata randomly assign one BRFSS health survey data record to donate their behaviour to each SEER cancer registry data record [9]. This latter approach is referred to as cold deck imputation [14], and is the method employed in the current study.

Some of the BRFSS health survey data records did not have complete data for all six of the health behaviour measures. To avoid replacing the missing behaviour with a missing value, six copies of the BRFSS data set were created, each containing complete data for one of the six behaviours. For each behaviour two BRFSS 'donor' records were randomly assigned, without replacement, to each SEER cancer registry case. Donor records had to be of the same sex, race, marital status and State of residence and had to be recorded 5 years earlier in time and be one 5-year age category younger than the SEER cancer registry case they were assigned to. As the required number of replications of the imputation process increases as the percentage of nonresponse gets larger [14], this cold deck imputation was repeated 100 times for each of the six health behaviours. SEER cancer registry cases with less than two eligible donor records were excluded from subsequent analyses on that behaviour. S1 Fig is a flow chart showing SEER cancer cases who were included and excluded from the analysis. S2 Fig is an equivalent flow chart for the BRFSS health survey data records. S1 Table describes the eligible SEER cancer registry cases and the proportion of these who were excluded due to unavailability of matching BRFSS health survey donor records for 'Current smoking'. Exclusions were higher in earlier years, older age groups, males, non-whites and Californians. Very similar results would be expected for the other five behaviours.

Any imputed data will contain errors, and imputation informed by demographic characteristics alone will contain many errors. However, it is known that people from similar demographic groups have a higher likelihood of having similar behaviour than people from different demographic groups [15–18]. If the behaviour of the SEER cancer case is donated by a demographically similar individual, then it should have a slightly higher likelihood of being correct than if it were obtained from a completely random donor. Given the selection of donors is random, the resulting misclassification errors will also be random. In statistics, random error is generally controlled through sample size. The more random error, the larger the sample size required to confidently detect the remaining signal amongst the random noise.

Sample size needs for the current study are investigated below. The requirement for a large data set may be the limiting factor when the disease is rare. However, the more variables in common and more informative those variables (stronger their relationship with the behaviour) the stronger the information signal [19] and the more likely it could be detected.

The effect of misclassification is to attenuate the results of the analysis towards the null (no effect) [20], but if the misclassification can be measured, it is possible to statistically correct this attenuation.

In the current study misclassification was estimated by imputing the behaviour twice and quantifying the disagreement between these two imputed values. The misclassification between the two imputed values is an estimate of the misclassification between the 'true' behaviour and the 'imputed' behaviour. That is, the agreement between behaviour between two individuals from the same demographic strata in BRFSS health survey was used as an estimate of the agreement in behaviour between an individual in the SEER cancer registry and an individual in the BRFSS health survey from the same demographic strata.

Of the 5 methods for misclassification correction for Cox regression models reviewed by Bang et al [21], the method used in the current study is corrected score estimation (as it is suitable for a dichotomous predictor, and external measure of the misclassification error) [22]. For each of the 100 repetitions of the imputations in the data set, the misclassification rate and fitted the misclassification corrected Cox regression model with corrected score estimation were separately calculated.

## Simulation method

As the true hazard ratios (HRs) are unknown, it was not possible to evaluate the effectiveness of the I2C2 algorithm using the real data alone. To test the algorithm, 100 copies of the SEER cancer registry data were simulated containing both a 'true' and an 'imputed' smoking status.

Starting with the 100 repetitions of the SEER cancer registry data set with imputed smoking status, censored data were excluded, leaving 23,657 data records in each data set. The proportion of cancer cases in each age group was recorded as well as the average proportion of imputed smokers and non-smokers in each age category, and the average agreement between the two smoking categories in each age category across the 100 data sets. Finally the distribution of observed survival times was compared to a range of probability models using the skewness-kurtosis plot generated by the 'descdist()' command in 'fitdistrplus' package in R software. In the absence of clear alternatives, the Weibull model was applied. Survival times following a Weibull distribution can satisfy the proportional hazards requirements of Cox regression while also allowing for any future investigation of alternate survival models.

Next 100 simulated data sets were created, each of which contained

- A code number of the data set

- Age group

- A smoking status representing the 'true' smoking status

- A smoking status representing the 'imputed' smoking status

- 14 simulated survival times

In each simulated data set, there were 23,657 data records. These were randomly assigned to age groups and 'true' and 'imputed' smoking status groups such that the proportion of cases in each age group, the proportion of smokers in each age category and the misclassification between the 'true' and 'imputed' smoking status were all approximately the same as in the real dataset. The simulated survival times were produced by the method of Bender et al [23] for simulating Weibull survival times with specified HRs, and then rounded to the nearest integer (months). The first seven survival times for each data record correspond to HRs of 0.50, 0.67, 0.80, 1.00, 1.25, 1.50 and 2.00 given 'true' smoking as the only predictor. The second set of seven HRs are calculated with both smoking and age category as predictors of survival status, with the HR for age set at the value in the SEER cancer registry data and the HRs for 'true' smoking at HRs of 0.50, 0.67, 0.80, 1.00, 1.25, 1.50 and 2.00 as above.

The first set of seven survival times are not confounded with age (as age is omitted in the development of the HRs) but the second set of survival times have the same level of confounding by age as the real data (as the proportion of smokers differed by age and both smoking status and age are predictors of survival).

## Statistical analysis

Each of the six health behaviours were recorded as dichotomous variables. Each of the 100 repetitions of the data set for each health behaviour contained two records of that health behaviour: two imputed values for the SEER cancer registry data and a designated 'true' and a designated 'imputed' measure in the simulations.

Each statistic was estimated in each of the 100 repetitions of the data set for each health behaviour. Results are presented as the median of these 100 observations and the corresponding 95% empirical confidence interval (CI); the 2.5 and 97.5 percentiles.

To summarise how much information had been retained through the imputation process, the level of agreement beyond chance between the two imputed values (or between the true and imputed value in the simulations) is reported. Both Cohen's Kappa and the direct calculation of the difference in the observed and expected number of people recorded as having the behaviour are reported on both imputations from the cross-tabulation of the two imputed values (or true against imputed values in the simulated data sets).

Analyses of survival time were conducted using Cox models using the Breslow method for addressing ties. For the simulated data the proportionality assumption was tested using the z-test on Schoenfeld residuals against transformed time. (Given the ideal distributions of the simulation, the median p-value was about 0.5.) Correction for misclassification was applied using the corrected scores estimation method. Results were presented as the median of the estimated HRs with associated 95% empirical CIs.

All analyses were conducted in R v4.0.2. The corrected score estimation software was provided by its creator, Prof David Zucker, as a Fortran 77 program which was called from within R using the foreign function interface (dyn.load() command). It was not possible to test the proportionality assumption when using corrected scores estimation and models were restricted to the Breslow method for addressing ties.

Information passed to the corrected score calculator included the data set and two misclassification rates were passed:

- the proportion positive on the first imputation (or true behaviour) who were coded as negative on the second imputation; and

- the proportion negative on the first imputation (or true behaviour) who were coded as positive on the second imputation; and

Often the imputation process provided insufficient information for the corrected scores estimation to be calculated. While there are a number of ways to check for insufficient information, in this study a pragmatic criteria was adopted in that whenever the estimated HRs tended towards positive or negative infinity the model was deemed to have failed (see S3 Fig). The number of failures in the 100 data sets are reported in bar charts. Where more than 5 out of 100 data sets return estimated HRs of <0.01 or greater than 100, that analysis was labelled as failed and the results were not reported. Where one to five data sets return extreme HRs, these extreme results were omitted from medians and 95% CIs.

## Results

Agreement beyond chance was highest for imputed 'smoking status' with a median Kappa of 0.07 and a median of 299 more smoker-to-smoker matches (and 299 more non-smoker to non-smoker matches) observed than expected through chance agreement alone (Table 1). This agreement beyond chance scaled to the sample size is estimated to be 0.009. The next highest rates of information transfer were observed for 'Physical activity' followed by 'Obesity' and 'Binge drinking'. The two least common behaviours–'Heavy drinking' and 'Smoking with regular alcohol' with median prevalence of 4.8% and 3.3% respectively–had virtually no information retained through the imputation process.

Fig 1 shows the effectiveness of the I2C2 algorithm when applied to simulated data with smoking status as sole predictor of survival time in Cox regression. The median HR obtained from the I2C2 estimation conforms quite well with the median of the true HRs but, as expected, the CIs arising from the I2C2 algorithm are much wider. The wider CIs reflect the additional random error arising from the misclassification errors in the imputation.

Fig 2 shows that the I2C2 algorithm fails when the relationship between smoking status and survival time is confounded by age. The 'effect' of smoking status (which is subject to high levels of misclassification error) shown in the first graph is being incorrectly attributed to age (which is measured without misclassification error) shown in the second graph.

Age confounding was addressed using both standardisation and stratification.

**Table 1. Selected statistics describing the agreement between the two donor records across the 100 repetitions of the SEER cancer registry data for each of the six behaviours.**

|  | Sample size | Proportion with behaviour | Kappa (95% CI) | Agreement beyond chance (number) | Agreement beyond chance (rate) |
|---|---|---|---|---|---|
|  |  | Median (95% CI) | Median (95% CI) | Median (95% CI) | Median (95% CI) |
| **Smoking status** | 31,754 | 15.7 (15.7,15.9) | 0.07 (0.06,0.08) | 299 (246,353) | 0.009 (0.008,0.011) |
| **Binge drinking** | 31,674 | 10.0 (9.9,10.2) | 0.06 (0.04,0.07) | 167 (124,203) | 0.005 (0.004,0.006) |
| **Heavy drinking** | 31,671 | 4.8 (4.7,5.0) | 0.01 (<0.01,0.02) | 16 (2,33) | <0.001 (<0.001,0.001) |
| **Physical activity** | 31,758 | 73.8 (73.6,74.1) | 0.03 (0.02,0.04) | 214 (145,273) | 0.007 (0.005,0.009) |
| **Obesity** | 31,718 | 26.1 (25.8,26.3) | 0.03 (0.02,0.04) | 194 (130,261) | 0.006 (0.004,0.008) |
| **Smoking with regular alcohol** | 31,653 | 3.3 (3.2,3.4) | 0.02 (0.01,0.04) | 22 (10,39) | <0.001 (<0.001,0.001) |
| **Simulated smoking status** | 23,657 | 15.8 (15.5,16.2) | 0.07 (0.06,0.09) | 236 (194,279) | 0.010 (0.008,0.012) |

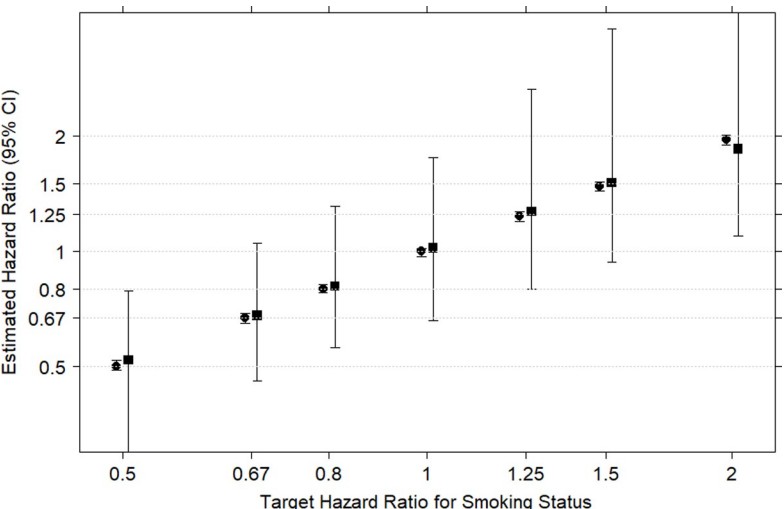

**Fig 1. Results from 100 simulated data sets each with the same sample size and proportion of smokers as the SEER cancer registry data.** Each simulated data set contains seven different survival times associated with the target HRs on the horizontal axis. The vertical axis shows median and 95% empirical CIs of estimated HRs arising from Cox models. The first CI in each pair is obtained from fitting designated 'true' smoking status as a predictor of survival using the standard Cox model. The second is obtained from fitting designated 'imputed' smoking status using Cox regression with corrected score estimation (the I2C2 algorithm).

All survival times were first standardised to the average age group. That is, the Cox regression predicting survival time was first fitted using age group alone. The observed survival times were then scaled by the predicted age effect (using the predict() option in R). Fig 3 shows that weight-based confounding adjustment works quite well for the designated 'true' smoking with little residual attenuation to the null. Unfortunately, this small attenuation becomes much larger when magnified by the high levels of misclassification error arising through the I2C2 algorithm. Fig 3 shows considerable residual attenuation of the median HR towards null effect (and, as expected, the much wider CIs for the estimated HRs arising from the misclassification errors) associated with 'imputed' smoking status after age standardisation.

Results from applying the I2C2 algorithm to the SEER cancer registry data with imputed behaviour are shown in in Fig 4. The bar chart shows that the algorithm has, as expected, often failed to converge on a HR estimate for 'Heavy drinking' and 'Smoking with alcohol' and has also failed for 'Obesity'. Therefore, results for these behaviours have been suppressed. The second chart suggests that smoking 5 years prior to diagnosis may be a hazard to survival (HR 1.32, 95% CI 0.82,2.68) but binge drinking could be protective (HR = 0.49, 95% CI 0.13,1.29). But both CIs include the null effect. Physical activity outside work is a statistically significantly protective of survival (HR = 0.25, 95% CI 0.03,0.81).

Equivalent analyses for the squamous cell carcinoma sub-group and the adenocarcinoma sub-group are presented in S4 and S5 Figs respectively. For most of these analyses there was insufficient information passed through the imputation process to allow misclassification correction to converge. 'Smoking status' was associated with a statistically significantly increase in hazard in the squamous cell carcinoma subgroup (HR 1.93, 95%CI 1.08–7.07), but appeared to have no association with the adenocarcinomas (HR 0.92, 95%CI 0.51–1.74), 'Physical activity' may be protective in squamous cell carcinoma (HR 0.34, 95%CI 0.07–1.64), but the results were not statistically significant, and 'Obesity' appears to have no relationship with survival time in squamous cell carcinoma (HR 1.08, 95%CI 0.22–4.96). The simulations suggest these results are likely underestimates.

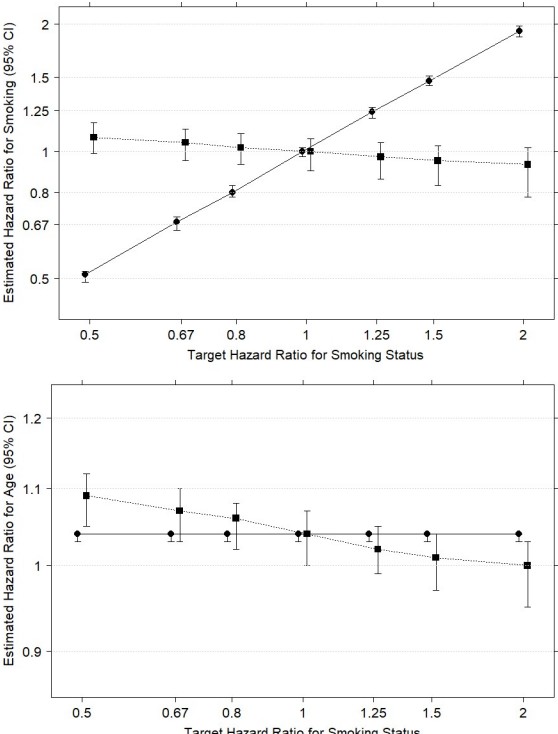

**Fig 2. Results from 100 simulated data sets containing age confounding.** The first chart shows estimated HRs for smoking status and the second shows estimated HRs for 5-year age group. The CIs with circles joined by solid lines are obtained from fitting the designated 'true' smoking status and 5-year age group as predictors of survival using standard Cox regression. The CIs with squares joined by dotted lines are obtained from fitting the designated 'imputed' smoking status using, adjusted for 5-year age category, using the I2C2 algorithm.

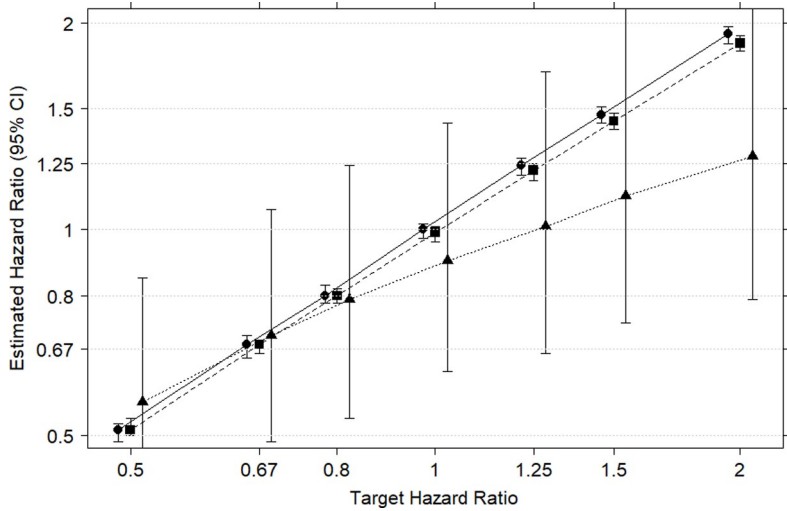

**Fig 3. Estimates of the HR of smoking 5 years prior to diagnosis on post-diagnosis survival from simulated data sets.** Three CIs are presented at each target HR. The first (denoted by circles joined by a solid line) are the estimated HRs obtained from Cox regression where 'true' smoking status and age are predictors of survival time. The second CIs (squares connected by a dashed line) shows the estimated HRs for 'true' smoking status after age-standardisation of survival times. The third set of CIs (triangles connected by a dotted line) shows the estimated HR for imputed smoking status after age standardisation (I2C2).

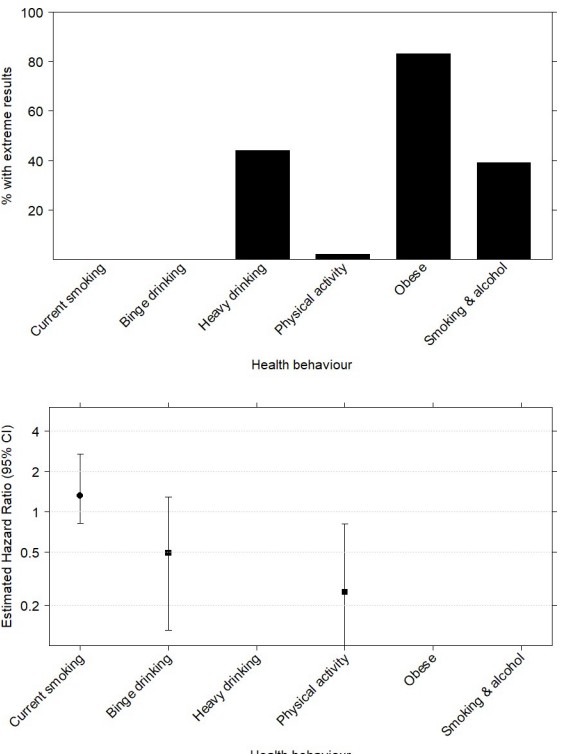

**Fig 4. A summary of the results obtained after age-standardisation of survival times.** The bar chart shows the number of data sets where the algorithm has returned extreme values (<0.01 or >100) for estimated HRs. The second chart shows the median estimated HR and associated 95% empirical CIs for those behaviours which have recorded 5 or less extreme HRs.

Using stratification to adjust for the confounding effect of age was also investigated, but the strata sample sizes proved to be too small for the I2C2 algorithm to produce interpretable results. For the simulated data sets for smoking status, S6 Fig shows that the corrected scores Cox regression often failed to converge and S7 Fig shows that even where results are available, the median HR from the I2C2 algorithm is a poor point estimate of the 'true' HR.

S8 Fig shows the relationship between sample size, agreement beyond chance rate and estimated HRs, using the prevalence of smoking from the impute values across the SEER cancer registry data sets. It shows that with simulated survival times (Weibull, no censoring), the I2C2 algorithm gives accurate point estimate (albeit with wide uncertainty) when agreement beyond chance as a rate is 0.008 or more and sample sizes are 20,000 or more. For agreement beyond chance of 0.006 sample sizes of 40,000 or 50,000 are required to produce accurate point estimates and for lower signal strengths the algorithm seems to generally do poorly. The available sample size of around 31,700 (including censored observations), stratified into 5-year age groups appears to be too small to support age stratified I2C2 analysis in the current example.

## Discussion

This study investigated the relationship between pre-diagnosis health behaviour and post-diagnosis oesophageal cancer survival by applying an imputation algorithm (the Impute, Impute, Calibrate, Correct, or I2C2, approach) to augment a cancer registry data set with information from an external, no-cases-in-common, health behaviour data set. Findings from

the study showed that, using simulated data without age confounding, the I2C2 algorithm provides an accurate estimate of the median HR for the predictor variable, albeit with wider CIs arising from the additional random error arising from misclassification.

In practice, despite encouraging results, the I2C2 algorithm is vulnerable to a number of problems. Sample size and information retention through the imputation process are important issues. In the current example, the I2C2 algorithm failed to produce any estimates for behaviours with low prevalence such as 'Heavy drinking' and 'Current smoking with regular drinking' (both with an estimated prevalence of <5% among the cancer cases) and in age groups with low prevalence such as 'Smoking status' among those 75 or more years of age.

Another issue was age confounding. Survival times decrease as age increases, and prevalence of many health behaviours also differ by age. For example, among the SEER cancer registry cases imputed smoking rates are lower in older age groups, while prevalence of obesity and physical activity outside of employment are both higher in older age groups. Age-standardisation was only partially successful at removing the age effect and sample sizes were insufficient for age-stratification.

The I2C2-based analyses did produce some suggestive results. Of the six behaviour measures studied, imputed smoking status 5-years prior to diagnosis appeared to retain the most information through the imputation with a median excess smoking-to-smoking matches of 299 (95% CI 246,353), or 0.009 (95% CI 0.008,0.011) of the sample. In the current study the age-standardised estimated median HR for smoking was 1.32 (95% CI 0.82,2.68). But previous studies have suggested the effect of smoking on survival may differ between sub-groups [12].

Recent meta analyses estimated HRs of 1.41 (95% CI 1.22,1.64) and 1.41 (95% CI 0.96,2.09) for current smoking relative to never smoked in mainly squamous cell populations [24, 25] with no evidence of association between smoking and survival in EAC [12, 25]. In the current study age-standardised results showed a statistically significantly increased HR in squamous cell carcinoma (HR = 1.93, 95%CI 1.08,7.07) with no apparent effect in adenocarcinoma (HR = 0.89, 95%CI 0.05,3.53). Based on the simulations, these results are likely to underestimate the true hazards.

Physical activity outside employment 5 years prior to diagnosis, recorded 214 (95% CI 145–273) more agreements between imputed values than predicted by chance. Physical activity appeared to be protective of survival age-standardised estimated HR of 0.25 (95%CI 0.03,0.81). A recent meta-analysis [26] combined results from a US and a Korean study and reported that pre-diagnosis physical activity to be protective of post-diagnosis survival in oesophageal cancer (HR = 0.77, 95%CI 0.59–1.00). No evidence was found for difference between squamous cell and adenocarcinoma in the association between physical activity outside work and survival.

I2C2-derived HR estimates for other health behaviours were sparse and in one case problematic: the tendency for 'binge drinking' 5 years prior to diagnosis to be protective of age-standardised survival (HR 0.49, 95% CI 0.13,1.29) may suggest a failure of age standardisation.

These findings suggest potential in the I2C2 algorithm, and with the benefit of experience, there are many ways to improve its performance. Sample size could be increased by including more years or addressing the unnecessary exclusions. The variables used are quite limited: each of the 6 behaviours were dichotomous and obviously dichotomous variables convey the least possible information about behaviour. Some of the demographic variables (such as 5-year age groups, State of residence, etc) could have conveyed more information if divided into smaller categories. The imputation process was parsimonious: simply excluding cancer registry cases with less than two donor records rather than attempt to find nearest neighbours etc. Model-based imputation [27] may be more informative than simple

donor records. Additional investigation of the measurement of misclassification would also strengthen the approach. For example, internal calibration, confirming the true health behaviour from a sub-set of the cancer cases could give a more direct measure of misclassification. Similarly, a quick and simple method for age-standardisation was used, which has been subject to previous criticism [28]. Ideally, it may be possible to refine the corrected scores estimation algorithm to address confounding variables directly (such as ability to specify offsets or fix the coefficient for age). Finally, the I2C2 needs to be more thoroughly tested. A prospective or retrospective cohort which contained true pre-diagnosis behaviours as well post-diagnosis survival times would allow gold standard comparisons between true the HR and I2C2-estimated HR.

The variety and size of health surveys and prospective cohorts will continue to increase over time. All such large human data sets tend to contain a good variety of demographic measures allowing us to differentiate people with similar and dissimilar risk factors. Unlike one-to-one matching, the I2C2 algorithm produces almost no additional confidentiality risk over and above the original cancer registry. (The data which is being added to the cancer registry is largely misclassifications.)

Further improvements in the I2C2 algorithm will permit enrichment of cancer registry data through imputation of new variables with negligible risk to patient confidentiality, opening new research opportunities in cancer epidemiology.

## Supporting information

**S1 Fig. Flow chart of inclusions and exclusions of SEER oesophageal cancer cases.**
(DOCX)

**S2 Fig. Flow chart of inclusions and exclusions of BRFSS health behaviour data records.**
(DOCX)

**S3 Fig. An example of Cox regression coefficient tending towards negative infinity.**
(DOCX)

**S4 Fig. Age-standardised hazard ratios for simulated smoking in squamous cell subgroup.**
(DOCX)

**S5 Fig. Age-standardised hazard ratios for simulated smoking in adenocarcinoma subgroup.**
(DOCX)

**S6 Fig. The proportion of data sets failing to converge (HR<0.01 or HR>100) when using I2C2 with age-stratified simulated smoking.**
(DOCX)

**S7 Fig. Age-stratified hazard ratios for simulated smoking in oesophageal cancer.**
(DOCX)

**S8 Fig. Simulation of the relationship between sample size, agreement beyond chance rate and estimated hazard ratios for smoking status.**
(DOCX)

**S1 Table. Number of SEER oesophageal cancer cases seeking donor records for current smoking behaviour and the proportion of these failing to obtain two donor records.**
(DOCX)

## Acknowledgments

The authors thank Prof David M. Zucker, Hebrew University, Jerusalem, Israel for sharing his computer program for corrected score estimation Cox Regression.

## Author Contributions

**Conceptualization:** Paul P. Fahey, Andrew Page, Thomas Astell-Burt, Glenn Stone.

**Data curation:** Paul P. Fahey.

**Formal analysis:** Paul P. Fahey, Glenn Stone.

**Methodology:** Paul P. Fahey, Andrew Page, Glenn Stone.

**Project administration:** Paul P. Fahey, Andrew Page.

**Software:** Paul P. Fahey, Glenn Stone.

**Supervision:** Paul P. Fahey, Andrew Page, Thomas Astell-Burt, Glenn Stone.

**Visualization:** Paul P. Fahey.

**Writing – original draft:** Paul P. Fahey.

**Writing – review & editing:** Paul P. Fahey, Andrew Page, Thomas Astell-Burt, Glenn Stone.

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
