## [Decision Letter · Decision Letter 0]

8 Nov 2021

PONE-D-21-16334

Imputing pre-diagnosis health behaviour in cancer registry data and investigating its relationship with oesophageal cancer survival time

PLOS ONE

Dear Dr. Paul Patrick Fahey,

Thank you for submitting your manuscript to PLOS ONE. After careful consideration, we feel that it has merit but does not fully meet PLOS ONE’s publication criteria as it currently stands. Therefore, we invite you to submit a revised version of the manuscript that addresses the points raised during the review process.

We look forward to receiving your revised manuscript.

Kind regards,

Surasak Saokaew, PharmD, PhD, BPHCP, FACP

Academic Editor

PLOS ONE

Journal Requirements:

Additional Editor Comments:

1. Individuals who survived beyond 2015 and those who were lost to follow-up had their censored survival time recorded at their last known survival date. Form this statement, how the author manages the censored: as negative outcome, positive outcome, or as dummy please clarify.

2. As we know that “The more random error, the larger the sample size required to confidently detect the remaining signal amongst the random noise”. In this case, how many samples are required for this study, please described and calculate.

3. Please specify the rationale why the author chose Weibull survival time. 

Reviewers' comments:

Reviewer's Responses to Questions

**Comments to the Author**

1. Is the manuscript technically sound, and do the data support the conclusions?

Reviewer #1: Yes

Reviewer #2: Yes

2. Has the statistical analysis been performed appropriately and rigorously? 

Reviewer #1: Yes

Reviewer #2: Yes

3. Have the authors made all data underlying the findings in their manuscript fully available?

Reviewer #1: Yes

Reviewer #2: No

4. Is the manuscript presented in an intelligible fashion and written in standard English?

Reviewer #1: Yes

Reviewer #2: Yes

5. Review Comments to the Author

Reviewer #1: The manuscript utilizes an innovative approach towards pre-diagnosis health behaviour among oesophageal cancer patients, and demonstrates the application of I2C2 algorithm quite well. In my point of view, this manuscript would be helpful in enriching cancer registry data across the country.

Reviewer #2: Reviewer comments

Abstract:

a. Unclear from where was data extracted and collected: There was computerized and valid data? Specify this?

b. In results: You should express measure of association such as HR, 95%CI

Introduction:

a. In line: example, 28, 60,101,174 and across the manuscript, please, avoids to write pronouns as”we”,..and instead it write nouns as"The study ....."

References:

a. Please, check format reference number 11 according to the Guidelines provided at the PLOSO ONE website

6. PLOS authors have the option to publish the peer review history of their article (what does this mean?). If published, this will include your full peer review and any attached files.

Reviewer #1: **Yes: **Dr. Shekhar Grover

Reviewer #2: No

---

## [Author Response · Author response to Decision Letter 0]

11 Nov 2021

Additional Editor Comments:

Individuals who survived beyond 2015 and those who were lost to follow-up had their censored survival time recorded at their last known survival date. Form this statement, how the author manages the censored: as negative outcome, positive outcome, or as dummy please clarify.

Response: We feel that the coding and analyse of censored observations are quite standardised and that further elaboration may be more confusing than informative. (That is, it may leave the reader asking if we have done something out of the ordinary.) In the models reported, a censored observation is included as alive at all time points up to censoring time and excluded from analyses for all time points after the censoring time. 

2. As we know that “The more random error, the larger the sample size required to confidently detect the remaining signal amongst the random noise”. In this case, how many samples are required for this study, please described and calculate.

Response: We have added the sentence “Sample size needs for the current study are investigated below.” The issue is addressed in final paragraph of the results and supplementary Figure 9.

3. Please specify the rationale why the author chose Weibull survival time. 

Response: We have added further details of this decision-making process. The updated text is:

“Finally the distribution of observed survival times was compared to a range of probability models using the skewness-kurtosis plot generated by the ‘descdist()’ command in ‘fitdistrplus’ package in R software. In the absence of clear alternatives, the Weibull model was applied. Survival times following a Weibull distribution can satisfy the proportional hazards requirements of Cox regression while also allowing for any future investigation of alternate survival models.”

Reviewer #1: The manuscript utilizes an innovative approach towards pre-diagnosis health behaviour among oesophageal cancer patients, and demonstrates the application of I2C2 algorithm quite well. In my point of view, this manuscript would be helpful in enriching cancer registry data across the country.

Response: Thank you.

Reviewer #2: Reviewer comments

Abstract:

a. Unclear from where was data extracted and collected: There was computerized and valid data? Specify this?

Response: We have added “Data from US cancer registries and a US national health survey are used in testing the algorithm.”

b. In results: You should express measure of association such as HR, 95%CI

Response: The reviewer may have missed that the Results section of the Abstract contains the description: “The estimate of the hazard ratio of pre-diagnosis smoking was 1.32 (95% CI 0.82,2.68). To make this clearer we have added “HR=” to each of the numeric results.

Introduction:

a. In line: example, 28, 60,101,174 and across the manuscript, please, avoids to write pronouns as”we”,..and instead it write nouns as"The study ....."

Response: We have removed all instances of “we” and “our”, except for a few references to our previous publications.

References:

a. Please, check format reference number 11 according to the Guidelines provided at the PLOSO ONE website

Response: We are using the PLoS style in Endnote. We have changed the description of the item from “Manuscript” to “Book” and believe that this has addressed the problem.

---

## [Editor Report · Decision Letter 1]

2 Dec 2021

Imputing pre-diagnosis health behaviour in cancer registry data and investigating its relationship with oesophageal cancer survival time

PONE-D-21-16334R1

Dear Dr. Paul Patrick Fahey,

We’re pleased to inform you that your manuscript has been judged scientifically suitable for publication and will be formally accepted for publication once it meets all outstanding technical requirements.

Kind regards,

Surasak Saokaew, PharmD, PhD, BPHCP, FACP

Academic Editor

PLOS ONE
---

## [Editor Report · Acceptance letter]

6 Dec 2021

PONE-D-21-16334R1 

Imputing pre-diagnosis health behaviour in cancer registry data and investigating its relationship with oesophageal cancer survival time 

Dear Dr. Fahey:

I'm pleased to inform you that your manuscript has been deemed suitable for publication in PLOS ONE. Congratulations! Your manuscript is now with our production department. 

Kind regards, 

on behalf of

Dr. Surasak Saokaew 

Academic Editor

PLOS ONE